Adaptive changes in bodybuilders in conditions of different energy supply modes and intensity of training load regimes using machine and free weight exercises

Chernozub Andrii 1
Manolachi Veaceslav manolachivsciences@yahoo.com 2 3
Tsos Anatolii 1
Potop Vladimir 3 4 5
Korobeynikov Georgiy 6 7
Manolachi Victor 2 3
Sherstiuk Liudmyla 8
Zhao Jie 6
Mihaila Ion 4 5
1 Lesya Ukrainka Volyn National University , Lutsk , Ukraine
2 Dunarea de Jos University of Galati , Galati , Romania
3 State University of Physical Education and Sport , Chisinau , Republic of Moldova
4 Department of Physical Education and Sport, University of Pitesti , Pitesti , Romania
5 Doctoral School of Sports Science and Physical Education, University of Pitesti , Pitesti , Romania
6 National University of Physical Education and Sport , Kyiv , Ukraine
7 Institute of Psychology, German Sport University Cologne , Cologne , Germany
8 Petro Mohyla Black Sea National University , Mikolaiv , Ukraine
Badicu Georgian
Electronic publication date: 2023 Feb 17
Publication date: 2023
Volume: 11
Electronic Location ID: e14878
Received 2022 Nov 15; Accepted 2023 Jan 20
Copyright: ©2023 Chernozub et al.
Copyright year: 2023
Copyright holder: Chernozub et al.
License: This is an open access article distributed under the terms of the Creative Commons Attribution License, which permits unrestricted use, distribution, reproduction and adaptation in any medium and for any purpose provided that it is properly attributed. For attribution, the original author(s), title, publication source (PeerJ) and either DOI or URL of the article must be cited.
License URL: https://creativecommons.org/licenses/by/4.0/

Keywords: Adaptation, Maximal strength, Anaerobic-alactate mode, Creatinine concentration, Fat-free mass, Workload, Athletes

Funding: The authors received no funding for this work.

==============================
Background

The research was aimed at comparing the effect of using two types of training load different in intensity and energy supply. We studied the influence of the proposed load variations (machine and free weight exercises) on long-term adaptation of the body at the stage of specialized basic training in bodybuilding.

Methods

A total of 64 athletes aged 18–20 years were examined. The research participants were randomly divided into four groups, 16 athletes in each group. Athletes of group 1 and 3 used a complex of free weight exercises. Group 2 and 4 participants performed machine exercises. Bodybuilders of group 1 and 2 were trained in conditions of medium intensity training load (Ra = 0.58) in the anaerobic-glycolytic mode of energy supply. Athletes of the 3rd and 4th groups used high intensity load (Ra = 0.71) in the anaerobic-alactate mode of energy supply. We managed to determine the nature of adaptation processes using methods of control testing of strength capabilities, bioimpedansometry, anthropometry, biochemical analysis of blood serum (LDH, creatinine).

Results

The study showed that the difference in the dynamics of the participants’ maximum strength development (on example of chest muscles) did not depend on the content of machine or free weight exercises, but on the features of training load regimes. Thus, the controlled indicator of strength capabilities in athletes of groups 3 and 4 increased by 5.1% compared to groups 1 and 2. During all stages of the study, the indicators of the projectile working mass in athletes of groups 3 and 4 exceeded the results observed in groups 1 and 2 by 25.9%. At the same time, the amount of load in a set is on average 2 times higher in athletes of groups 1 and 2. Group 4 athletes, who used machine exercises and high intensity training load, increased the circumferential body measurements by 3.8 times (the chest), compared to the results recorded in group 1 athletes. Athletes of group 1 and 2 showed increasing in body fat by 3.4% compared to the initial level on the background of large load volume. The basal creatine level in bodybuilders of groups 3 and 4 increased by 3.7 times after 12 weeks of study, which indicates an accelerated growth of muscle mass.

Conclusion

The most pronounced adaptive body changes in bodybuilders at the stage of specialized basic training occurred during high intensity training load and anaerobic-alactate energy supply mode. Machine exercises contributed to increasing the morpho functional indicators of athletes more than free weight exercises.

Introduction

Improving the training process by optimizing loads is one of the main issues in bodybuilding. There is a constant search for effective combination of strength exercises and training programs. The issue of adaptation to a stressful stimulus in bodybuilding remains relevant. Adaptation changes are of great interest among leading specialists (Aerenhouts & D’Hondt, 2020; Coratella et al., 2020; Heidel, Novak & Dankel, 2022). This problem is especially acute at the stage of specialized basic training in bodybuilding. The complexity of the process of training optimization is associated with a high level of body resistance to physical exertion. Even if the load parameters contribute to the accelerated growth of muscle mass and strength (Chernozub et al., 2020; Marshall et al., 2020; Schwanbeck et al., 2020). At the same time, using a wide range of means, principles and training methods in the previous stages complicates the mechanisms of increasing adaptation reserves in the shortest possible time and requires the search for new ways of improving the training process (Chernozub et al., 2018; Saeterbakken et al., 2022).

The impact of training loads different in volume and intensity on adaptation processes in untrained people is of steady interest among scientists. Scientists in sports physiology and fitness have been paying attention to the research of this problem in recent years (Titova et al., 2018; Saeterbakken et al., 2019; Schott, Johnen & Holfelder, 2019). The integral method of quantitative estimation of load capacity in power fitness (Chernozub et al., 2020; Chernozub et al., 2022) allowed to assess power load parameters which depend on the muscle activity conditions and the level of functional capabilities. The load factor (Ra) reflects the indicator of the load regime intensity in power sports. This coefficient was estimated with the help of maximum strength indicators, work duration in a set, movement amplitude, duration of concentric and eccentric phases, and number of repetitions in a set. Indicators of projectile working mass (m) and load volume in the set (Wn) demonstrate the magnitude of the physical stimulus depending on the training load regime. The effectiveness of using quantitative indicators for assessing the intensity of load regimes was studied in fitness and MMA (Chernozub et al., 2020; Chernozub et al., 2022).

The complex use of morphometric and physiological indicators allows to evaluate the effectiveness of training loads and their adequacy to body capabilities. Blood biochemical markers helped to determine the nature of adaptive and compensatory reactions to a physical stimulus in different regimes of training loads. At the same time, the control of biochemical blood markers shows the level of resistance to training loads in certain modes of energy supply. It is known that the growth of maximum strength and muscle mass occurs after loads in anaerobic modes of energy supply (alactate and glycolytic). These changes occur due to hypertrophy of fast-twitch muscle fibers, growth of intra-muscular and inter-muscular coordination. The alactate and glycolytic anaerobic modes of energy supply differ in the mechanism of action, sources, duration and power (Shaner et al., 2014; Parks et al., 2020; Wilke, Stricker & Usedly, 2020; Wilk, Zajac & Tufano, 2021). At the same time, the question of finding an effective combination of training load (machine or with free weight exercises) in bodybuilding that differ in terms of energy supply and intensity has not been investigated by scientists yet.

For many years, the question of determining the effectiveness of using machine or free weight exercises in bodybuilding, fitness, powerlifting has caused conflicting views among scientists, coaches and athletes (Wirth et al., 2016; Wilke, Stricker & Usedly, 2020). In recent years, a number of specialists have paid close attention to the study of this issue not only in athletes, but also in people who use complexes of similar strength exercises in the process of muscle activity during rehabilitation (Schott, Johnen & Holfelder, 2019; Marshall et al., 2020). At the same time, the issue of determining the nature of changes in the concentration of testosterone, somatropin, and cortisol in the blood serum in response to training during machine and free weight exercises has been actively investigated (Shaner et al., 2014; Schwanbeck et al., 2020; González-Hernández et al., 2022). The results of these studies contribute to the in-depth observation of the features of body adaptive and compensatory reactions to physical stimuli when using similar complexes of training exercises (Stajer, Vranes & Ostojic, 2018; Chernozub et al., 2020; Saeterbakken et al., 2022). However, despite the sufficient number of studies in this direction, there is no clear definition of informative biochemical blood markers for assessing the growth of muscle mass, increasing the functional capabilities of bodybuilders.

Thus, the main problem of bodybuilding is the lack of physiologically based mechanisms for the ratio of external stimulus indicators and adaptation processes. Another pressing issue at the stage of specialized basic training is the effective combination of training means and load modes because athletes have high level of resistance to power loads. The in-depth study of these problems will allow to improve the training process in bodybuilding.

The purpose of this study was to determine the impact of training loads different in intensity and energy supply on performance in bodybuilding; to study the peculiarities of adaptive body changes using machine and free weight exercises at the stage of specialized basic training.

Materials & Methods

Participants

A total of 64 athletes aged 18–20 years were examined. Their experience of training in bodybuilding is 4.2 ± 0.4 years. A series of studies was conducted over 12 weeks in 2021 on the basis of fitness centers Gold Gym, Fight House, Septem Fitness, and Gym Style (Ukraine). The research participants were randomly divided into four groups, 16 athletes in each group. Athletes of groups 1 and 3 used a complex training regime which included free weight exercises. Participants of groups 2 and 4 used machine exercises for all training. However, bodybuilders of groups 1 and 2 trained in conditions of medium intensity training load (Ra = 0.58) in the anaerobic-glycolytic mode of energy supply. Athletes of the 3rd and 4th groups used high intensity training load (Ra = 0.71) in the anaerobic-alactate mode of energy supply.

The algorithm, structure and methods of the study were approved by the ethical committee for biomedical research of Lesya Ukrainka Volyn National University (ecbr18.08.2022) in accordance with the ethical standards of the Declaration of Helsinki. Participants gave written informed consent for the research in accordance with the recommendations of the biomedical research ethics committees (WHO, 2000).

Diagnostic equipment of the university medical and biological laboratory was used for conducting medical examination, assessment of functional status, indicators of body composition and biochemical control of creatinine concentration and LDH activity in the blood serum of the participants of the examined groups.

Maximal muscle strength

Monitoring the development of chest muscles maximum strength (1 RM) in the examined bodybuilders took place in four stages: at the beginning of the study and every 4 weeks. The control exercise, “barbell press lying on a horizontal bench”, was used to determine the maximum strength of the chest muscles of athletes performing free weight exercises. To determine the chest muscles maximum strength, we used the control exercise “bench press on the Smith simulator”. The procedure of control testing of the studied indicator 1 RM was carried out according to the generally accepted methodology for power sports with the observance of the appropriate exercise technique (e.g., Loturco et al., 2021).

Training load parameters

Using the integral method of quantitative estimation of load capacity in power fitness (e.g., Chernozub et al., 2018) was determined the load factor (Ra) at the beginning of the study. Its parameters reflect the intensity level of the power load regime. At each of the four stages of control (at the beginning of the study and after every 4 weeks), average group indicators of projectile working mass (m) and load volume (Wn) were determined in each set.

Circumferential body measurements

Body circumference measurements (on the example of chest) were made using a centimeter tape following the generally accepted methodology. Controlled indicators were measured at the beginning and during the next 12 weeks of the study with an interval of 30 days.

Body composition

Determination of body composition indicators was carried out using the method of bio electrical impedance (BIA) followed by computer processing of the obtained results. Body composition parameters were recorded at the beginning and during the next 12 weeks of the study with an interval of 30 days. The method of BIA was used to control the content of fat-free mass (FFM, kg) and body fat (BF) in the study participants. The KM-AR-01 Diamond-AST diagnostic computerized hardware and software complex (VYUSK. 941118.001 RE) (e.g., Martyrosov, Nikolayev & Rudnev, 2006) was used to determine the studied indicators of body composition.

Biochemical parameters

The creatinine concentration and the activity of lactate dehydrogenase (LDH) in the blood serum of bodybuilders were determined by the kinetic method on the equipment of the company “High Technology Inc” (USA) with a set of reagents PRESTIGE 24i LQ LDH (Poland). Blood sampling was performed by a medical worker in compliance with the internationally accepted requirements for medical and biological research (e.g., Tietz, Finley & Pruden, 1995). Control of changes in the parameters of the studied biochemical blood indicators took place at the beginning and after 12 weeks of training at rest (before exercise) and at the end of the training session.

Experimental design

The research was conducted in several stages:

At the first stage, machine and free weight exercises were selected to achieve the study objective. These complexes consisted of strength training exercises, which optimally engage the working muscle groups and allow to achieve maximum muscle fatigue in the required period of time. The machine exercises (using the example of basic exercises) were the following: Smith machine bench press, head off bench press, block thrust behind the head, standing cinder block triceps extension, Larry Scott curl, lying leg press. The free weight exercises (on the example of basic exercises) included: barbell chest press, seated barbell (dumbbell) bench press, bent-over row, French bench press, cross body hammer curl, barbell squat. Using the integral method of quantitative estimation of load capacity in power fitness (e.g., Chernozub et al., 2018), load modes of high (Ra = 0.71) and medium (Ra = 0.58) intensity were developed. Taking into account the duration of motor activity until complete fatigue of working muscle groups in a separate set, training load were performed in conditions of anaerobic-alactate or anaerobic-glycolytic modes of energy supply. During 12 weeks of research, 36 training sessions were conducted. The training duration was 30-33 min. Not more than 3–4 muscle groups were loaded during one training. We used 1 basic and 2 isolated exercises for each group of muscles.

At the second stage, we studied the peculiarities of changes in maximum muscle strength development (using the example of chest muscles), circumferential body measurements (the chest), parameters of body composition (BF and FFM), indicators of projectile working mass (m) and load volume (Wn) in a set. The analysis of the studied morpho functional indicators allowed to clearly determine the nature of adaptive body changes in bodybuilders at this stage of training in conditions of different energy supply and intensity of training load regimes using machine and free weight exercises. We studied the character of changes in lactate dehydrogenase activity and creatinine concentration in the blood serum of athletes in response to a stress stimulus (training load) in order to determine the nature of the adaptive and compensatory body reactions.

At the third stage, the results of circumference measurements, indicators of fat-free mass and body fat were compared during 12 weeks of using the proposed machine and free weight exercises in conditions of different energy supply modes and intensity training load regimes, according to the nature of changes in creatinine basal level. We compared the results of changes in the concentration of LDH activity in the blood serum of athletes who used anaerobic-glycolytic and anaerobic-alactate modes of energy supply for muscle activity. The obtained results will low to determine the optimal combination of using machine or free weight exercises on the background of high (Ra = 0.71) or medium (Ra = 0.58) training load intensity, which will contribute to the most effective increase in functional capabilities and muscle mass growth in bodybuilders at the stage of specialized basic training.

Statistical analysis

Statistical analysis of the research results was performed using the IBM *SPSS*Statistics 26 program package (StatSoftInc., USA). Median, lower and upper quartiles, interquartile range (IQR) were determined. Kruskal-Wallis H test was used for testing whether samples originate from the same distribution. Friedman’s two-way analysis of variance by ranks was used to compare indicators of the same sample of subjects during the control period. Kendall’s W (the Kendall’s coefficient of concordance) is an effect size index for Friedman test. The Kolmogorov–Smirnov test was used to determine the normal distribution (e.g., Nasledov, 2013). The G-Power 3.1.96 program was used to calculate statistical power (determining the smallest sample size for the study). The sample size was evaluated using statistical tests: Wilcoxon signet-rank test (one sample case); ANOVA: repeated measures, between factors. Statistical results are reported according to the APA style (e.g., Yağin et al., 2021).

Results

Table 1 presents the dynamics of the chest muscles maximum strength development (1 RM) in athletes of all four groups during 12 weeks of the study.

Table 1 Dynamics of the chest muscles maximum strength development (1 RM) in bodybuilders of the examined groups during the study (IQR), n = 64.

Strength exercises	Groups	Term of observation, weeks	χ2, df = 3
W	
		Initial data	4	8	12		
Barbell chest press (kg)	1	113.50 (23.25)
U1= 105.00
p = 0.39	120.50
(23.27)6.2%1,*	127.00
(21.75)5.4%1,*	131.00
(22.50)3.1%1,*
15.4%2,***	χ2 = 46.87***
W = 0.97***	
3	115.00 (20.00)
U1= 105.00
p = 0.39	124.25
(21.88)8.0%1,*	132.00
(23.00)6.2%1,*	137.50
(19.75)4.1%1,*
19.5%2,***	χ2 = 48.00***
W = 1.00***	
Smith machine bench press (kg)	2	111.00 (16.75)
U2=124.00 p = 0.89	118.00
(17.75)6.3%1,*	123.00
(16.12)4.2%1,*	127.75
(16.87)3.8%1,*
15.1%2,***	χ2 = 48.00***
W = 1.00***	
4	110.00 (4.37)
U2= 124.00 p = 0.89	119.50
(5.37)8.6%1,*	127.75
(5.87)6.9%1,*	133.50
(3.00)4.5%1,*
21.3%2,***	χ2 = 48.00***
W = 1.00***	
Notes.

1 difference (%) compared to previous results.

2 difference (%) in comparison with the initial data.

df is the number of degrees of freedom

U1 Mann-Whitney test (groups 1 and 3)

U2 Mann-Whitney test (groups 2 and 4)

χ2 Friedman’s test

W is the Kendall’s coefficient of concordance

* p < .05.

*** p < .001.

Table 1 shows that during the period of research, there was positive dynamics in the chest muscles maximum strength development in bodybuilders of all groups. The most pronounced growth of the studied indicator was found in athletes of the 4th group (by 21.3%), who used machine exercises with high intensity training load in conditions of anaerobic-alactate mode of energy supply. The smallest development of the chest muscle maximum strength was observed in group 2 athletes (by 15.1%) using free weight exercises in conditions of medium intensity training load and anaerobic-glycolytic mode of energy supply. At the same time, in groups of athletes in the conditions of using training load identical in intensity, but different sets of exercises, the average-group difference between the dynamics of the studied indicator is not significant.

Training load parameters

The results of the average group changes in the parameters of projectile working mass, used by bodybuilders during 12 weeks of the study, are graphically presented in Fig. 1.

Figure 1 demonstrates that despite almost identical parameters of the maximum muscle strength development in all groups at the beginning of the study (Table 1), the controlled indicators of the projectile working mass was by 22.7% higher in athletes of groups 3 and 4 who used high intensity training load (Ra = 0.71) regardless of the type of exercises (Fig. 1). After 12 weeks of training, the bodybuilders of groups 3 and 4 showed a similar difference in the projectile working mass parameters compared to group 1 and 2 athletes, who used medium intensity training load (Ra = 0.58).

Figure 2 depicts the results of average group changes in the load volume in a set. Due to the peculiarities of medium-intensity training load mode, group 1 and 2 athletes used 2 times higher volume of training load from the very beginning of the study. The last 4 weeks of research showed that group 1 and 3 representatives, who used the same type of free weight exercises on the background of different load modes, had a 175.4% difference between the groups in the load volume in a set. However, group 2 and 4 athletes, who used machine exercises in conditions of medium (Ra = 0.58) and high (Ra = 0.71) intensity training load, had an identical difference in the studied indicator.

Figure 1 Dynamics of the average group indicators of the projectile working mass (m), used by all study participants during 12 weeks of research.

Figure 2 Dynamics of the average group indicators of load volume (Wn) in a set, recorded in the examined groups during 3 months of research.

Circumferential body measurements

The dynamics of the chest circumference measurements in the bodybuilders of examined groups is shown in Table 2.

Table 2 Changes in the chest circumference measurements in bodybuilders of the examined groups during the study (IQR), n = 64.

Groups	Term of observation, weeks	χ2, df = 3, W	
	Initial data	4	8	12		
1	109.00 (3.12)
H = 9.69, p = 0.07	108.15 (2.75)
−0,8%1,*	109.25 (2.88)
1.0%1,*	110.40 (2.30)
1.0%1,*
1.3%2,***	χ2 = 37.75***
W = 0.78***	
3	107.00 (1.28)
H = 9.69, p = 0.07	108.80 (1.30)
1,7%1,*	110.15 (1.15)
1.2%1,*	111.15 (0.87)
0.9%1,*
3.9%2,***	χ2 = 48.00***
W = 1.00*,***	
2	107.35 (4.75)
H = 9.69, p = 0.07	107.00 (4.65)
−0,31	108.20 (4.33)
1.1%1,*	110.50 (4.57)
2.1%1,*
2.9%2,***	χ2 = 45.99***
W = 0.95***	
4	109.50 (2.87)
H = 9.69, p = 0.07	112.50 (1.98)
2,7%1,*	113.85 (1.20)
1.2%1,*	114.85 (1.02)
0.9%1,*
4.9%2,***	χ2 = 47.71***
W = 0.99***	
Notes.

1 difference (%) compared to previous results.

2 difference (%) in comparison with the initial data.

df is the number of degrees of freedom

H Kruskal-Wallis H test

χ2 Friedman’s test

W is the Kendall’s coefficient of concordance

* p < .05.

*** p < .001.

Table 2 clearly depicts that chest circumferential measurements were positively growing in athletes of all 4 groups during the study. But the largest increase in the studied indicator among the examined bodybuilders was in group 4 athletes (by 4.9%) using machine exercises with high intensity training load in the anaerobic-alactate mode of energy supply. At the same time, group 1 athletes, who used free weight exercises in conditions of anaerobic-glycolytic mode of energy supply and medium-intensity training load (Ra = 0.58), showed an increase in the studied indicator by 1.3% for a similar period of time, comparing with participants of other groups.

Body composition

The changes in the indicators of fat-free mass and body fat of athletes who took part in 12-week research are given in Table 3.

Table 3 Results of changes in body fat and fat-free mass (kg) of bodybuilders of the examined groups during the study (IQR), n = 64.

Groups	Terms of observation, weeks	χ2, df = 3, W	
	Initial data	4	8	12		
Fat-free mass, kg	
1	72.43 (5.86)
H = 1.78, p = 0.61	73.65 (9.52)
1,7%1,*	74.11 (8,30)
0,6%1,*	74.93 (8.41)
1.1%1,*, 3.4%2,***	χ2 = 28.58***
W = 0.59***	
3	71.10 (6.88)
H = 1.78, p = 0.61	73.20 (7.39)
2,9%1,**	73.70 (6.86)
0,7%1,*	73.83 (7,01)
0.2%, 3.8%2,***	χ2 = 38.77***
W = 0.80***	
2	71.14 (7.63)
H = 1.78, p = 0.61	71.42 (7.41)
0,4%	72.59 (8.51)
1,6%1*	73.39 (7.86)
1.1%1, 3.1%2,***	χ2 = 29.10***
W = 0.60***	
4	73.08 (10.51)
H = 1.78, p = 0.61	75.23 (9.74)
2,9%1,*	76.67 (9.00)
1,9%1,*	77.29 (8.05)
0.8%1, 5.7%2,***	χ2 = 45.07***
W = 0.93***	
Body fat, %	
1	15.43 (3.91)
H = 1.71, p = 0.63	13.11 (3.07)
−2,3%1,*	12.02 (1.65)
−1,1%1,*	11.39 (1.42)
−0.6%, −4.0%2,***	χ2 = 45.90***
W = 0.95***	
3	14.45 (1.78)
H = 1.71, p = 0.63	14.34 (1.46)
−0.11	14.38 (1.48)
0,041	14.34 (1.52)
−0.041, −0.1%2	χ2 = 14.38***
W = 0.30***	
2	15.98 (3.33)
H = 1.71, p = 0.63	13.46 (1.60)
−2,5%1,*	12.60 (1.67)
−0,9%1,*	12.33 (1.74)
−0.3%, −3.6%2,***	χ2 = 41.18***
W = 0.85***	
4	15.32 (3.55)
H = 1.71, p = 0.63	14.51 (2.28)
−0.8%1	14.43 (1.75)
−0.11	14.65 (1.67)
0.21, −0.6%2	χ2 = 4.72
W = 0.09	
Notes.

1 difference (%) compared to previous results.

2 difference (%) in comparison with the initial data.

df is the number of degrees of freedom

H Kruskal-Wallis H test

χ2 Friedman’s test

W is the Kendall’s coefficient of concordance

* p < .05.

*** p < .001.

The results of BIA showed that body fat index decreased greatly (by 4.0%) in group 1 athletes compared to the initial data. The index of fat-free mass in the same group of bodybuilders increased by 3.1% compared to the results recorded in representatives of other groups.

The fat-free mass index in group 4 athletes showed the largest increase of 5.7% among the examined bodybuilders over a period of 12 weeks of training. However, their body fat index did not change significantly over the same period of time.

Biochemical parameters

Figure 3 graphically presents the changes in lactate dehydrogenase activity in the blood serum of all group participants using complexes machine and free weight exercises during all stages of the research.

Figure 3 Changes in LDH activity in the blood serum of athletes of the examined groups using free weight exercises (A) and machine exercises (B) during the study, n = 64.

1. compared to the indicators before exercises; *p < .05.

Figure 3 shows that, LDH activity increased in the blood serum of bodybuilders in the following way: group 1—by 16.6%, and group 2—by 14.5%, compared to the state of rest. It happened in response to a stress stimulus at the beginning of the study. However, the activity of this enzyme in group 3 and 4 athletes in response to high intensity training load during machine exercises did not change significantly.

After 12 weeks of training, LDH activity in the blood serum increased with a smaller progression, and only in athletes of group 1 (by 13.1%) and 2 (by 6.8%) in response to medium intensity training load in anaerobic-glycolytic energy supply mode. At the same time, the studied enzyme activity in group 3 and 4 athletes did not change in response to training load in the conditions of anaerobic-alactate energy supply mode, regardless of using the type of strength exercises.

Figure 4 displays the results of changes in the creatinine concentration in the blood serum of the examined bodybuilders in response to different modes of anaerobic energy supply performing machine and free weight exercises.

Figure 4 The change in creatinine concentration in the blood serum of athletes of the examined groups using free weight exercises (A) and machine exercises (B) during the study, n = 64.

1. compared to the indicators before exercise; 2. compared to the indicators before the study; *p < .05.

The results analysis shows that the initial measurements of creatinine concentration in the blood serum decreased in athletes of all groups in response to a stress stimulus. Thus, in representatives of groups 3 and 4, the concentration of the studied biochemical indicator shows an average 2-fold decrease in response to high intensity training load in the conditions of anaerobic-alactate energy supply mode, compared to the results recorded in bodybuilders of groups 1 and 2. At the same time, there was no significant difference between the effect of using machine or free weight exercises on the parameters of the studied biochemical indicator.

After 12 weeks of training, the creatinine concentration demonstrated changes in athletes of all examined groups in response to a stress stimulus. Thus, its concentration increased by 7.6% (<0.05) on average in athletes of all groups. The basal creatinine level was 3.7 times higher in athletes of groups 3 and 4. These changes in the basal level of creatinine indicate an accelerated growth of muscle mass in athletes of groups 3 and 4.

Discussion

This research describes the influence of training loads of different intensity and energy supply on performance in bodybuilding. It studies the process of adaptation to training loads (machine and free weight exercises) at the stage of specialized basic training. The study results indicate that there is no significant difference between machine and free weight exercises at this stage of training. We have identified that it is the high-intensity loads in combination with the anaerobic-alactate mode of energy supply that contributed to maximum adaptive body changes. The results of this study will contribute to the optimization of training loads at the stage of specialized basic training. The obtained results will also help to better understand the processes of adaptation in bodybuilding while using high and medium intensity training loads on the background of anaerobic modes of energy supply.

The lack of uniform mechanisms for training activity optimization is one of the unsolved problems in the modern training system of bodybuilding. The combination of effective load variations, principles, methods and tools is one of the most controversial scientific directions. In-depth study of adaptive body changes in bodybuilders is one of the priority areas of research in sports physiology (Tang, Chan & Kuo, 2014; Chernozub et al., 2020; Saeterbakken et al., 2022). Determining the peculiarities of adaptive body changes of bodybuilders using machine and free weight exercises in different modes of energy supply and their effectiveness is one of the directions of this problem. However, the majority of studies are specifically aimed at the comparative assessment of the effectiveness of using machine and with free weight exercises due to the dynamics of morpho functional indicators and changes in certain biochemical blood indicators (Wochyński & Sobiech, 2017; Aerenhouts & D’Hondt, 2020; Wilke, Stricker & Usedly, 2020). At the same time, leading scientists in this direction paid only partial attention to the course of adaptive changes in conditions of using these exercises on the background of different intensity of training load and modes of anaerobic energy supply allowing to increase the effectiveness in the shortest possible time while reducing the risk of adaptation failure.

The greatest increase in maximum muscle strength and circumferential measurements of bodybuilders occurred during high-intensity training loads (Ra = 0.71) in the anaerobic-alactate mode of energy supply. The rate of morpho functional indicators growth does not depend on type of exercises (machine or free weight). These adaptive changes are associated with accelerated hypertrophy of fast-twitch type B muscle fibers, which actively participate during exercises in anaerobic-alactate mode of energy supply (Joanisse et al., 2020; Lopez et al., 2021). At the same time, the increase in the maximum muscle strength indicators is associated with growing indicators of intra-muscular and inter-muscular coordination in conditions of the anaerobic-alactate mode of energy supply (Wilk, Zajac & Tufano, 2021; Carvalho et al., 2022).

The results of bioimpedance measurement indicate that using free weight exercises and medium intensity training load significantly decreased the level of body fat with a minimal increase in fat-free mass. Such changes in this body composition indicator highlights significant energy expenditure due to the large amount of work performed and the use of a significant number of stabilizer muscle groups during free weight exercises, as well as the possible manifestation of compensatory reactions to a physical stimulus (e.g., Chernozub et al., 2020). The level of body fat of athletes performing machine exercises in the anaerobic-alactate mode of energy supply did not change during the research. However, the fat-free mass increased two times compared to the results of the participants using anaerobic-glycolytic energy supply mode for the same period, which indicates long-term adaptation on the background of pronounced hypertrophy of fast-twitch muscle fibers (e.g., Moesgaard et al., 2022).

The indicator of the projectile working mass was 25% higher in conditions of high intensity training load, and the parameters of load volume in a series of four sets were twice smaller compared to the results found during medium intensity training load despite identical initial parameters for the development of maximum muscle strength of the examined groups. At the same time, the use of machine or free-weight exercises did not affect the parameters of these load indicators. The characteristic difference in the projectile working mass values and performed volume load depends entirely on the power load regime and the variability of the parameters of its main components. (e.g., Chernozub et al., 2018). Thus, doubling the duration of concentric and especially eccentric phases of movement significantly reduces not only the level of inertia during exercise, but also the number of repetitions in a set, which is necessary to ensure the fatigue of the working muscles within the limits of the predominantly anaerobic-alactate mode of energy supply (Chernozub et al., 2020; Kubo, Ikebukuro & Yata, 2021).

The increase in lactate dehydrogenase activity in the blood serum of bodybuilders in response to medium-intensity training load are confirmed by the results of a series of studies in power fitness (e.g., Titova et al., 2018). A significant increase in LDH activity within the physiological norm in response to strength training indicates the accumulation of lactate in the blood of athletes during muscle activity and the activation of the predominantly anaerobic-glycolytic mechanism of ATP resynthesis (e.g., Wochyński & Sobiech, 2017). The level of LDH activity in response to high intensity training load during strength exercises, especially machine exercises, did not change both in conditions of short-term and long-term adaptation. This is due to the fact that in these conditions of muscle activity, energy supply occurs due to the creatine phosphokinase mechanism of ATP resynthesis and the absence of lactate in the blood (Stajer, Vranes & Ostojic, 2018; González-Hernández et al., 2022). The obtained results also indicated that there was no need to involve additional muscle groups during machine exercises, which are extremely necessary for stabilizing the body position in free weight exercises (e.g., Chernozub et al., 2018).

The changes in creatinine concentration did not depend on the type of exercises. Rather, the change of this blood biochemical marker depended up on the intensity of loads and the mode of energy supply. The obtained data confirm the results of research in fitness (e.g., Chernozub et al., 2020). However, the increase in the basal level of creatinine concentration by 6 times indicate an increase in the adaptive reserves for the creatine phosphokinase mechanism of ATP resynthesis during anaerobic-alactate mode of energy supply and high intensity training load (e.g., Tang, Chan & Kuo, 2014).

Thus, the obtained results indicate the expediency of using high intensity load (Ra = 0.71) and anaerobic-alactate energy supply mode in combination with machine exercises at the stage of specialized basic training in bodybuilding. The study proved that adaptive body changes were the most pronounced on the background of accelerated growth of strength and muscle mass with minimal reduction in body fat (Coratella et al., 2020; Heidel, Novak & Dankel, 2022). Biochemical indicators of LDH activity and creatinine concentration in the blood serum are not only markers for evaluating adaptive body changes in conditions of strength load of various intensity (Stajer, Vranes & Ostojic, 2018; Chernozub et al., 2020; González-Hernández et al., 2022), but they also contribute to the development of new mechanisms for improving the training process. The optimization of training process takes place due to the correction of load volume components taking into account those processes that will take place in the body depending on the involvement of certain mechanisms of energy supply.

Conclusions

Using sets of machine exercises in combination with high intensity training load contribute to the growth of strength capabilities and muscle mass of bodybuilders at the stage of specialized basic training in the shortest possible time. The absence of additional involvement of stabilizer muscles in the process of performing machine exercises allows to reduce energy consumption in most cases, and using high intensity training load in the conditions of anaerobic-alactate energy supply mode will allow to increase the reserves of the creatine phosphokinase mechanism of ATP resynthesis and thus the efficiency of the long-term adaptation process as a whole. The implementation of the study results in the training activity will help to find effective ways of optimizing training load taking into account the adaptation potential of the body, the peculiarities of the training stages and muscle activity.

The studied blood biochemical markers were used for evaluating adaptive changes in bodybuilders in response to high and medium intensity training load while using machine and free weight exercises. A clear distinction of changes in LDH activity and creatinine concentration in the blood serum in response to this stress stimulus allows to determine the level of involvement of certain energy supply modes and energy recovery mechanisms during muscle activity, which in the perspective will help with developing innovative systems for load correction and optimization of the training process as a whole.

Supplemental Information

Supplemental Information 1 Determine the nature of adaptation processes using methods of control testing of strength capabilities, bioimpedansometry, anthropometry, biochemical analysis of blood serum (LDH, creatinine)

(a) Initial data. (b) 4 weeks. (c) 8 weeks and (d) 12 weeks. LDH - lactate dehydrogenase. (A) free weight exercises. (B) machine exercises (B). FFM - fat-free mass (kg). BF - body fat (%).

The dynamics of the chest muscle’s maximum strength development (1 RM) in athletes of all 4 groups during 12 weeks of the study; the results of average group changes in the load volume in a set; the dynamics of the chest circumference measurements in the bodybuilders of examined groups; the changes in the indicators of fat-free mass and body fat of athletes who took part in 12-week research; the changes in lactate dehydrogenase activity in the blood serum of all group participants using complexes machine and free weight exercises during all stages of the research and the results of changes in the creatinine concentration in the blood serum of the examined bodybuilders in response to different modes of anaerobic energy supply performing machine and free weight exercises.

Click here for additional data file.

The article is a part of the planned scientific work, “Development and implementation of innovative technologies and correction of the functional state of a person during physical activity in sports and physical therapy”, (state registration number 0117U007145) and the Ministry of Education and Science of Ukraine project number 0118U000809.

Additional Information and Declarations

Competing Interests

Author Contributions

Human Ethics

Data Availability

The authors declare there are no competing interests.

Andrii Chernozub conceived and designed the experiments, performed the experiments, analyzed the data, prepared figures and/or tables, authored or reviewed drafts of the article, and approved the final draft.

Veaceslav Manolachi conceived and designed the experiments, performed the experiments, analyzed the data, prepared figures and/or tables, authored or reviewed drafts of the article, and approved the final draft.

Anatolii Tsos conceived and designed the experiments, analyzed the data, prepared figures and/or tables, authored or reviewed drafts of the article, and approved the final draft.

Vladimir Potop conceived and designed the experiments, performed the experiments, analyzed the data, prepared figures and/or tables, authored or reviewed drafts of the article, and approved the final draft.

Georgiy Korobeynikov conceived and designed the experiments, performed the experiments, analyzed the data, authored or reviewed drafts of the article, and approved the final draft.

Victor Manolachi conceived and designed the experiments, analyzed the data, prepared figures and/or tables, authored or reviewed drafts of the article, and approved the final draft.

Liudmyla Sherstiuk conceived and designed the experiments, performed the experiments, analyzed the data, prepared figures and/or tables, and approved the final draft.

Jie Zhao conceived and designed the experiments, performed the experiments, analyzed the data, authored or reviewed drafts of the article, and approved the final draft.

Ion Mihaila conceived and designed the experiments, analyzed the data, prepared figures and/or tables, authored or reviewed drafts of the article, and approved the final draft.

The following information was supplied relating to ethical approvals (i.e., approving body and any reference numbers):

The algorithm, structure and methods of the study were approved by the ethical committee for biomedical research of Lesya Ukrainka Volyn National University (ecbr18.08.2022).

The following information was supplied regarding data availability:

The raw measurements are available in the Supplemental File.

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
