# Peer review of "Adaptive changes in bodybuilders in conditions of different energy supply modes and intensity of training load regimes using machine and free weight exercises"

_PeerJ, doi:10.7717/peerj.14878_

## Round 0.1 · original submission · Major Revisions

Some major changes are needed to the article. Please address these changes and resubmit.

Best regards,

Georgian Badicu

·

Basic reporting

Minor revision are required in the English spelling.

Experimental design

The design of the study was well planned.

Validity of the findings

The results should be revised as in the items I mentioned.

1. How was the sample size determined? Power analysis should be included in the text.
2. I don't think your statistical approach is correct. First of all, you wrote that you used the Wilcoxon test, but you gave the average values. Wilcoxon is a non-parametric test and it is more appropriate to give the median and interquartile range (IQR) in descriptive statistics for data that are not normally distributed. In addition, although the design of the study was two-sided, you handled the analyzes one-way. My suggestion is to use repeated two-way ANOVA if your data meets the parametric test assumptions, and PERMANOVA if not. In this way, you can consolidate your results by examining the time*group interaction effect. Finally, it is important to present the results regarding the effect sizes and percentage change in tables.
3. The Summary, Results and discussion section should be rearranged following the correct analysis.
4. I suggest that you interpret your statistical results according to the APA style and add the relevant source by writing that the results are reported according to the APA style in the statistical analysis section. https://doi.org/10.52876/jcs.916182

Additional comments

If the revisions I suggest are made, the subject of the article is original and acceptable.

·

Basic reporting

The limitations and strengths of the study should be briefly presented within the discussion section. Additionally, it is highly recommended that an English specialist (or fluent English speaker) proofread the entire manuscript for grammar and readability.

Experimental design

This original research is nicely presented throughout this manuscript. However, there are some points to note.
The authors did define the research objective.

Validity of the findings

According to the analyzed research, the available evidence and findings will fill the existing gap in the knowledge in the field.

Additional comments

Ln102-104 “To determine the peculiarities…” – This sentence needs to be rephrased.
Ln104 “Complexes” – Please consider rephrasing assets/group. (Please check elsewhere in the manuscript for the same term and make corrections)
Ln105 “bodybuilding” – Do you mean bodybuilders? Please rephrase accordingly.
Ln164 “solve” - Please consider using another term here.

Ln175 “load” – Loads. Plural.

Ln267 “reliably” – Do you mean significantly?

Ln278 "at the beginning of the study" – This could also be rephrased as baseline measurements/ initial measurements. Consider rephrasing this part.

Ln288 “fixed” – Please check if this term is suitable. Please consider rephrasing if needed.

Ln289 “reliable” – Do you mean significant?

Ln291-294 "The study showed…." - This sentence is unclear. Please rephrase it.

Ln296-300 "In the modern…" - The sentence is too long, and please rephrase it to make it more understandable.

Ln323-325. Please check the grammar and rephrase the sentence.

Ln361 “changes” – changed.

Ln361-364. I suggest rephrasing the sentence.

Ln368 “course of studies” – What is meant by this particularly? Please recheck the meaning and rephrase if needed.

Ln388 “on the whole” – Please consider rephrasing as a whole.

Ln389 “will let us” – what exactly does this stand for? Please recheck the sentence carefully and rephrase it in a way that makes the point clearer.
Ln391 “condition” - What type of condition particularly?

Ln393 – “The studied by us biochemical indicators of blood are used as markers” – This part needs to be rephrased.

Ln398 “it” - This can be deleted.

Ln399 “on the whole” – Please consider rephrasing as “as a whole."

·

Basic reporting

Authors reported on an interesting study on adaptive changes in bodybuilders in conditions of different energy supply models and intensity of training load regimes using machine and free weight exercises. The manuscript is generally well-written, but there is a lack of info in introduction, rationale, and methodology. Given the importance of the subject in the sport science field, the manuscript should give more info on the matter. Although the study results are promising and important, I have some major methodological concerns and other issues that the authors need to address before I can accept the manuscript for publication.

Specific comments

Abstract
• The info should be better presented as to raise the interest of the readers. The more data the authors give, the more difficult the abstract is to read and the interest to be risen. There is not a real background, the two phrases are in fact the aims of the study.
• The key words must be different from the title words.

Introduction

• The information given by the authors is not sufficient to create the background for this important matter to sports field. Can the authors maybe provide some more background on different energy supply models and training load regimes?

• The rationale for examining this problem should be mentioned more clearly in this section. Why did the authors choose to examine it? Why didn’t the authors choose to study some other aspects?

• The article innovation should be presented in the Introduction. Describe what the research gap of the paper is and what is new. Please describe the links between the research gap and the goal of the article.

• Lines 72-74 contain too many” of”, which make it difficult to read and understand. This is the case for many parts of the article (for example, lines 296-300 and many other like those) as well as using long, sentences/phrases which could easily bore the reader. Research articles usually do not use the word "we/our/us" (lines 49, 164, 180, 393) and regularly use passive verbs.
I think the problem is the lack of proper familiarity with the English language and recommend that the entire article should be corrected by someone who is a native speaker/editor.

Experimental design

• It is necessary to specify if there was a priori analysis performed to establish the sample size. This information is essential to assume the soundness of the obtained results.

• The methodology is explained on the three stages the authors used in the study. An explanatory model or a diagram would be more useful to illustrate the features.

Validity of the findings

• The Results section should be reorganised as to follow each purpose. Authors need to write key findings focusing on each one of these to clearly present the findings.

• The Discussion section - first paragraph should refer to what is new regarding the findings of the study.

• It would be better to have seen more use of terms like 'originality' and 'significance'. Identify what is new in this study that may benefit readers or how it may advance existing knowledge or create new knowledge on this subject. There should be a clear conclusion on why the research findings are significant for this subject and could be of use for athletes in this situation.

• Research limitations and existing problems are not presented.

---

## Round 0.2 · Minor Revisions

Some minor changes are needed.

Best regards,

·

Basic reporting

The results of the current research will fill the knowledge gap in the field and minor revisions are required. The abstract, results and discussion are well-revised.

Experimental design

The experimental design of this research is well presented.

Validity of the findings

The authors improved their analysis and validated the results of the article. I recommend doing the power analysis (that is, the minimum number of athletes to be included in the research) using the G-Power program and adding it to the text.

Also check that the p value must be italicized throughout the text.

Additional comments

The revisions from the previous review have been taken into account. It can be adopted after existing recommendations are made.

Congratulations to the authors.

·

Basic reporting

The limitations and strengths of the study should be briefly presented within the discussion section.

Experimental design

This original research is nicely presented throughout this manuscript.
The authors did define the research objective.

Validity of the findings

According to the analyzed research, the available evidence and findings will fill the existing gap in the knowledge in the field.

Additional comments

The manuscript is now improved. I suggest accepting the manuscript.

·

Basic reporting

Thank you for providing this comprehensive work.
The authors have presented an improved version of the manuscript.

The introduction provides a proper background of the topic. The sections have been improved. Relevant results are well-organized to follow the hypothesis/each purpose.
The quality of the images is good enough.
The limitations and strengths of the study should be briefly presented within the discussion section. Additionally, it is highly recommended that an English specialist (or fluent English speaker) proofread the entire manuscript for grammar and readability.There are so many” of”-s in just one sentence or phrase. For example, lines 67-70 (of is mentioned 4 times), lines 74-78 (7 times), 79-81 (6 times) and so on. The use of the same word is also not advisable: line 60 “constant search” and line 72 “constant interest”.

Experimental design

The experimental design meets the scope of the journal, and it is relevant to the community.
Methods are described detailed enough.

Validity of the findings

The results and the conclusions are quite interesting and well-discussed. All data are provided.

---

## Round 0.3 · Minor Revisions

Some minor changes are needed. Please see the attached file from the Section Editot.

Best regards,

---

## Round 0.4 · Minor Revisions

The Authors misunderstood a comment from the Section Editor. As they have relied upon/indicated that the groups were randomly allocated. Sentences 133-134 should read "the research participants were randomly divided into 4 groups ". Once this minor edit is made I recommend the manuscript for publication.

With kind regards,
Georgian Badicu
Academic Editor
PeerJ Life & Environment

---

## Round 0.5 · accepted · Accept

Accepted for publication. Congratulations!